# Smart RFID Sensors Embedded in Building Structures for Early Damage Detection and Long-Term Monitoring

**DOI:** 10.3390/s19245514

**Published:** 2019-12-13

**Authors:** Christoph Strangfeld, Sergej Johann, Matthias Bartholmai

**Affiliations:** Bundesanstalt für Materialforschung und -prüfung, Unter den Eichen 87, 12205 Berlin, Germany; sergej.johann@bam.de (S.J.); matthias.bartholmai@bam.de (M.B.)

**Keywords:** RFID based sensors, embedded sensors, corresponding relative humidity, porous building materials, reinforced concrete, corrosion, civil engineering

## Abstract

In civil engineering, many structures are made of reinforced concrete. Most degradation processes relevant to this material, e.g., corrosion, are related to an increased level of material moisture. Therefore, moisture monitoring in reinforced concrete is regarded as a crucial method for structural health monitoring. In this study, passive radio frequency identification (RFID)-based sensors are embedded into the concrete. They are well suited for long-term operation over decades and are well protected against harsh environmental conditions. The energy supply and the data transfer of the humidity sensors are provided by RFID. The sensor casing materials are optimised to withstand the high alkaline environment in concrete, having pH values of more than 12. Membrane materials are also investigated to identify materials capable of enabling water vapour transport from the porous cement matrix to the embedded humidity sensor. By measuring the corresponding relative humidity with embedded passive RFID-based sensors, the cement hydration is monitored for 170 days. Moreover, long-term moisture monitoring is performed for more than 1000 days. The experiments show that embedded passive RFID-based sensors are highly suitable for long-term structural health monitoring in civil engineering.

## 1. Introduction

A safe and reliable infrastructure is the fundamental base for social coexistence and economic growth in our modern society. In civil engineering, investigated objects such as tunnels, bridges, foundations, and ship locks have projected lifespans of several decades up to more than 100 years. Aging and degradation processes of steel-reinforced concrete structures require periodic maintenance and servicing activities. The cumulative yearly investment in maintenance in the transport infrastructure amounts to ca. 13 billion euro in Germany [1]. Furthermore, approximately 71% of all infrastructure degradation is caused by chloride-induced corrosion or by carbonation-induced corrosion [2]. An additional 5% of this degradation is directly related to moisture processes during the freeze–thaw cycle [2]. On the other hand, road freight transport will increase by approximately 40% in Germany between 2010 and 2030 [3]. This will lead to significantly increased loads, which will accelerate the degradation of existing infrastructure. Similar problems are also known all around the globe, such as Europe, Asia, Africa, and North America. For example, in the United States of America the annual costs due to corrosion of highway bridges, as estimated in 2002, amounted to more than 8 billion [4]. 

To maintain our ageing infrastructure in good condition, intensive and efficient repair is required. Consequently, it should be emphasised that the basis of any efficient repair activity should be a detailed damage diagnosis and assessment. Nevertheless, it is difficult to obtain the required information for distinct damage mechanisms, especially if the damage emanates from inside the concrete structure. For example, chloride-induced corrosion occurs on the surface of reinforcing bars in steel-reinforced concrete. The initiation of this process cannot be detected by traditional visual inspections [5]. Rather, it is typically only after years or even decades of degradation that obvious damage can be observed on the surface of a structural component, i.e., cracks, rust stains, delamination of the concrete cover, etc. At this point, massive damage is already present, and the corresponding repair work is expensive. Early detection of the onset of corrosion reduces costs, ensures building safety, and may increase the lifetime of the structure [6,7,8]. Thus, proper monitoring of crucial infrastructure like bridges, pavements, and foundations will reduce the overall costs of our infrastructure significantly. 

As a complement to visual inspections, sensors are capable of acquiring further information. Such sensors are particularly useful for collecting information from locations that are difficult to access, such as foundations, bridge bearings, and the bottom side of bridge decks, as well as locations where access is prohibited or dangerous, such as in nuclear waste containments. Appropriate sensors measure several quantities, which give a deep insight into the structural condition. To advance the state-of-the-art in this crucial field the “Bundesanstalt für Materialforschung und -prüfung” (BAM, Federal Institute for Materials Research and Testing) develops sensors designed for civil engineering structures. In the current study, the focus is on the detection of increased moisture in the porous cement matrix. For the chemical reaction of corrosion to occur, a certain moisture level is required. In the case of “healthy” concrete with a corresponding relative humidity below 65%, the high ohmic resistance of the electrolytical coupling significantly decelerates or even completely hinders the corrosion process [9]. Thus, moisture is considered as a key parameter for monitoring in civil engineering [8,10]. Other studies have already tried to quantify the humidity of the porous cement matrix based on the measurement of the signal strength at varying frequencies [11,12,13,14] or by direct embedment of humidity sensors [15]. 

Although monitoring can help to reduce the high repair costs, it is a challenging task. The systems must be robust and reliable. Traditional sensors need cables for energy supply and communication. In the case of large structures, such as tunnels or bridges, the total length of cables rapidly amounts to several kilometres. This increases costs for implementation and maintenance significantly and often makes monitoring uneconomic [16]. Furthermore, cables weaken the concrete surface and represent a high risk for infiltration of moisture, carbon dioxide, or chloride ions. This might lead to the initiation of degradation processes near the sensors, even though the rest of the structure is in good condition. Battery-driven sensors and communication systems might overcome this issue, but they must generally be mounted on the surface of structures, potentially exposing them to harsh environmental conditions, vandalism, or theft. In contrast, concrete structures are well suited for embedded sensors and systems, but the projected lifetime for these structures is typically in excess of 30 years. Thus, embedded battery-driven systems are not generally feasible for service life monitoring. 

Based on these exclusion criteria, only passive systems remain as a suitable solution for monitoring over wide areas. In recent years, ordinary radio frequency identification (RFID) tags have been developed to RFID-based sensors in several applications [17,18]. The approach investigated in this study is based on wireless RFID-based sensors, which are embedded completely in concrete. High-frequency (HF) and ultra-high frequency (UHF) RFID are able to provide both wireless energy supply and communication. Thus, these systems are of great interest for monitoring in civil engineering [19,20]. The RFID-based sensors are passive and robust, capable of recording data for several decades, and detect moisture or corrosion at its initial state. Completely embedded sensors keep the concrete cover intact: by eliminating the need for openings or any other intrusions that could weaken the structure. They also require minimal installation effort. Non-destructive evaluation of building conditions based on accumulated RFID-based sensor data is a robust and reliable way of monitoring structures and further contributes to an efficient maintenance of our infrastructure [21,22]. 

This paper describes the measurement principle and the system design of an exemplary passive RFID-based sensor [23]. It also presents comprehensive results of validation experiments, especially regarding the challenges and potential for long-term application. Due to the alkaline environment and the long service time of structures, sensors have to be tested in realistic long-term experiments. The optimisation of RFID-based sensors for long-term functionality and robustness is the focus of the current study.

## 2. Passive Radio Frequency Identification (RFID)-Based Sensors for Embedment into Concrete

The principle of measuring the equilibrium moisture content based on embedded relative humidity sensors, the so called corresponding relative humidity, is described. Then, the adaption of the sensor to the RFID chip for communication and energy supply is discussed. At the end, the developed RFID based sensors are shown.

### 2.1. Measurement Principle of Passive RFID-Based Sensors for Civil Engineering

Reinforced concrete consists of three main components, the cement paste, the aggregates, and the reinforcement. In most degradation processes, the aggregates are considered as inert. The cement paste forms the cement matrix with its open porosity. Via the water layer in the partially saturated pores, chlorides are transported and chemical reactions occur, e.g., carbonation [24]. If the reinforcing bars depassivate, corrosion is initiated. Iron ions are dissolved and Iron(III) oxide-hydroxide is formed. 

In a partially saturated pore system, moisture transport is a two-phase flow. It is a combination of water transport in vapour phase and in liquid phase. The equilibrium between these two phases is described by the sorption isotherm. Thus, moisture can be quantified via the liquid or the vapour phase. In liquid phase, the moisture is quantified by the mass of moisture related to the dry mass of the sample. In vapour phase, the moisture is quantified in terms of relative humidity. In the following, we quantify the moisture by means of the vapour phase. Thus, the two expressions “material moisture” and “relative humidity” are considered as synonyms.

At a material moisture below 75% relative humidity (RH), the corrosion rate is very low and can be neglected. Above approximately 80% RH, the corrosion rate increases drastically. The corrosion process leads to a volume increase of the reinforcing bars, which in turn leads to micro-cracking and concrete spalling. Furthermore, the stress-bearing cross-section is reduced, which endangers the structural integrity. However, in general, carbonation and chloride penetration start from the surface and migrate into the concrete. Thus, the first layer of reinforcing bars has the highest risk of corrosion. The concrete cover separating these reinforcing bars from the environment is typically 3 cm to 7 cm in depth. RFID-based sensors are ideally suited for structural-health monitoring because they can be installed directly above the location where two reinforcing bars intersect and acquire measurements within a range of approximately 5 cm. 

The embedded sensors must face an aggressive environment. The annual temperature variations generally range between −20 °C and 60 °C. Even more challenging is the high pH-value in concrete which lies between 12 and 14. The high pH-value causes chemical degradation of several materials. Therefore, the circuit board and the sensors must be protected by a robust casing. Nevertheless, a connection to the environment is required to monitor the moisture, e.g., the corresponding relative humidity of the porous material. Figure 1 illustrates the approach. The capacitive humidity sensor is encapsulated in its casing. To measure humidity variations of the surrounding building material, a permeable membrane has to be built in to enable water vapour diffusion. On the one hand, this membrane should be hydrophobic to avoid any liquid water transport due to capillary suction. On the other hand, the water vapour permeability should be sufficient to ensure equalisation of the relative humidity through the membrane. For this to happen, the air volume inside the casing between the sensor and the membrane must be able to interact with the air volume of the porous cement matrix. A disequilibrium of the relative humidity, e.g., the water vapour partial pressure, leads to a diffusive moisture transport through the membrane, such that the relative humidity in the sensor casing comes to correspond to the relative humidity of the pore system. 

### 2.2. Layout and Design of the RFID Circuit

The challenges of designing and developing RFID-based sensors for permanent embedment are primarily related to the power supply, accessibility, and robustness. RFID is divided into two categories, near field and far field. In the near field, both low and high frequencies (LF and HF, respectively) are used, commonly ranging from 125 kHz to 13.56 MHz. In the far field, ultra-high frequencies (UHF) and microwaves (MW) are used, commonly ranging from 800 MHz to GHz. Regulations stipulate which frequency can be used in which areas [25]. 

Our research focuses primarily on HF (13.56 MHz, worldwide standard) and UHF (868 MHz, standardized within the EU) in order to explore and compare the feasibility of using each of these frequency ranges. Three components are required to use this technology: 1. the reader, which controls the energy and data transmission, 2. antennas at both the transmitter and the transponder to establish the coupling, and 3. the transponder, which prepares and makes available the necessary data. The transponder can be operated either actively or passively. In this study, the passive operating mode was used to develop a sensor system that is as durable as possible so that structural monitoring will be feasible over decades.

The available range between the reading antenna and the transponder antenna also depends on the coupling. In the near field, maximum reading ranges in air from 20 cm to 1.5 m are possible. In the far field, ranges from 3 m up to 300 m are achievable. The maximum reading range also highly depends on the operating mode and environment.

The advantages of near-field technology are the low sensitivity to material moisture and low susceptibility to disturbances caused by small conductive objects. Although the far-field approach enables large reading ranges, moisture on the surface and in the material strongly reflects and absorbs the electromagnetic waves. Direct contact or attachment to metallic surfaces, e.g., reinforcing bars, must be avoided for both technologies. 

The embedded sensor systems developed for this study consist of five components. One of these components is an analogue humidity sensor (Honeywell (Morristown, NJ, USA) HIH-5030), which is connected to the RFID integrated circuit (IC) with an internal temperature sensor (ams (Premstätten, Austria) SL13A for HF and ams SL900A for UHF). The analogue output signal of the humidity sensor has been adapted to the requirements of the analogue-to-digital converter (ADC) of the RFID IC by means of a matching circuit (Figure 2), which has been configured in accordance with the superpositions principle (Equation1). The humidity sensor is supplied with energy generated by the energy harvester (EHV), which is capable of supplying up to 3 V, is regulated with the voltage regulator (REF193 (analog devices, Norwood, MA, USA)), and is attached to an antenna that is capable of meeting the capacitance requirements of the analogue front end (AFE) (Figure 3) (3.3 pF at HF and 39 nH at UHF). Further details about the electrical schematics can be found in [26,27].
U_Sig_ = R_x_/R_y_ (U_Offset_ − U_sig_) + R_x_/R_z_ (U_Sens_ − U_Sig_)(1)

With a corresponding encapsulation (see Section 4.3), the RFID transponders were embedded in test specimens in order to investigate their long-term stability. Figure 4 provides a photograph of the HF RFID-based sensor and Figure 5 shows the UHF RFID based sensor. Both sensor systems are based on commercial components (see description above). The overall system concept with the corresponding HF and UHF circuits, however, is a proprietary development of the BAM.

Each technology (HF or UHF) requires a special antenna. HF RFID transponders require a loop antenna to establish the coupling. In contrast, UHF RFID transponders use a dipole antenna. The following antenna geometries were used for the RFID transponders in this project: planar antennas consisting of 35 µm copper thickness, which are mounted directly on the carrier board and fulfil the requirements of flame-retardant class 4 (FR4).

## 3. Experimental Setup

First, the used humidity sensor is discussed, followed by a description of the hardware to evaluate the signal strength of the passive RFID-based sensors. Then, the used climate chambers for testing the specimens are described. Eventually, the setup of the exposure tests to sodium hydroxide (SH) solution is summarised. 

### 3.1. Humidity Sensors

A detailed description of the embedded humidity sensors is already available [24]. The accuracy of the HIH-5030 sensors is given as declared by the manufacturer. Within the range of 11% RH ≤ h ≤ 89% RH, the accuracy is ±3% RH, including interchangeability. Below 11% RH and above 89% RH, the accuracy amounts to ±7% RH including a reversible hysteresis of 3% RH. The overall repeatability for one single sensor is declared with 0.5% RH. These given numbers appear to be fairly high. Thus, in addition, five HIH-5030 sensors (Honeywell, Morristown, NJ, USA) of the same batch are tested for a more precise characterisation of the accuracy. These five sensors are calibrated by the dew point hygrometer S4000 Climatic of the company Michell Instruments (Friedrichsdorf, Germany). This reference hygrometer operates at ambient pressure and a temperature of 23 °C. The gas mass cell of the hygrometer itself is placed in a climate chamber [28] and has a gas flow of 1 lmin^−1^. Three succeeding cycles with eight humidity steps between 30% ≤ RH ≤ 95% are performed. In total, 225 individual humidity measurements are recorded, and the overall maximum deviation is 1.0% RH. The averaged deviation is approximately 0.4% RH and the mean correlation coefficient of the recorded measurement values to a linear function is r^2^ = 0.999. Assuming these five sensors are a representative test sample for the identical embedded humidity sensors, the overall accuracy is approximately ±0.5% RH [28].

### 3.2. Determination of the Signal Strength

The used Scemtec (Reichshof-Wehnrath, Germany) SAT-A4-LR-PP antenna with TI (Dallas, TX, USA) TRF7960A EVM transceiver reads out the embedded HF and the Meshed Systems (Ottobrunn, Germany) UHF patch antenna with ams (Premstätten, Austria) AS3993 Radon transceiver UHF sensors. The transmitter is positioned directly on the surface of the cement mortar specimen. The wooden template encompasses the mortar specimen and enables reliable and aligned transmitter positioning. The estimated positioning variance is approx. ±1 mm. Further details are given in [29]. In preliminary tests, the position with the lowest transmitting power is evaluated experimentally with hole spacing of 1 cm. The Voyantic Tagformance (Espoo, Finland) reveals the RFID tag performance by measuring the transmit power and the magnetic field strength. For HF, the frequency band is set to 13–14 MHz with a step width of 0.01 MHz. For UHF, the frequency band is 860 MHz to 900 MHz with 1 MHz steps. The minimum transmit power step was selected for the two frequencies, i.e., 0.25 dBm steps for HF and 0.1 dBm steps for UHF.

### 3.3. Determination of the Specimen Weight and Storage

Each specimen is weighted before measuring the humidity of the embedded sensors and the signal strength. A high-precision balance with a maximum load of 72 kg and an accuracy of 0.1 g is used as a reference measurement system. In that way, the empty weights of the casing and the embedded sensors are subtracted. These gravimetric measurements give a precise and reliable reference of the averaged water content of each specimen. 

The specimens are stored in a ventilated climate chamber. The size of this climate chamber is approximately 2.5 m × 2.5 m × 2 m. The specimens never left the 19 mm thick, wooden formwork. The bottom is a polyethylene foil, which is fixed at the formwork by means of tacker staples. This setup should ensure that evaporation only occurs on the top surface of the specimens. The ambient relative humidity is set to 50% RH and the ambient temperature is 23 °C. During the first summer period from days 168 to 233, the regulation was not able to hold the 50% RH and an increased humidity of approximately 54% RH to 60% RH is observed. However, the regulation of the climate chamber failed during the second and third summer period, as can be seen in Figure 6. Ambient humidity above 70% RH is observed between measurement days 461 to 739 and from 962 to 987. After day 987, the specimen was stored in another laboratory. Furthermore, after day 962, the ambient temperature decreased to 15 °C. A variation of the ambient humidity and temperature will affect the measurement values of the RFID-based sensors as well. This is discussed in Section 4.3.

### 3.4. Test of the Casing Materials in Sodium Hydroxide (SH) Solution

All casing materials are tested in an SH solution. For the experiment, 10 g of dried SH is solved in 250 mL distilled water. The resulting pH (SH) value is 12.7 at a temperature of 25.3 °C. As a first step, all specimens are dried in an oven at 60 °C for two hours. All weights are measured by a high precision balance with an accuracy of 0.1 mg. This procedure was carried out on all five measurement days. The entire test lasted 12 days. During that time, the pH value varied between 12.45 and 12.83 and the temperature between 21.2 °C and 25.4 °C. 

## 4. Results and Interpretation 

First, measurement results of the hydration of different cements are shown. This was a preliminary case study for the long-term tests. The measurement results of the long-term tests are discussed subsequently. From both tests, sensors were extracted from the concrete specimen and further investigated. The condition of the casing materials and the humidity sensor elements is discussed. In addition, results of material tests in a one molar SH solution are shown. An overview is given in Table 1. 

### 4.1. Preliminary Case Study of RFID-Based Sensors Embedded in Concrete

In a first step, prototypes of the RFID based sensors were tested in fresh concrete to evaluate the measurement setup and functionality. In this preliminary study, the casing material of all RFID-based sensors is made of epoxy resin, while in subsequent studies PVC cased sensors (Figure 3 and Figure 4) were investigated. HF sensors were embedded in different types of concrete in a preliminary test. The specific parameters of the cement pastes are depicted in Table 2. The sensors were embedded at 3 cm, 6 cm, and 9 cm depths. The total height of the concrete specimen is 10 cm. 

Figure 7 shows the corresponding relative humidity measured by the embedded HF sensors at 3 cm depth. The first promising result of this preliminary study is that the embedded HF sensors worked as expected. Communication and energy supply were good enough to measure a realistic trend of the humidity value. After 172 days, the specimens were cracked open to extract and analyse the embedded sensor components and to investigate the possible degradation of the sensor materials and components.

The main purpose of the preliminary study was to prove of the measurement concept. However, some more details can be seen in the measurement data. The total specimen weight of C1 is illustrated as well in Figure 7. It starts at 21.14 kg and decreases to 20.86 kg. Thus, approximately 0.3 L of water has left the specimen because of evaporation. Nevertheless, the main portion of the mixing-water added during concreting is hydrated. In this process the liquid water is chemically bound in the cement matrix, e.g., in the main product, calcium silicate hydrate. When the mixing-water hydrates and evaporates, the material moisture and the corresponding relative humidity decrease. As shown in Figure 7, during the first measurement days, the measured relative humidity is 100% RH. This means that the pore system is oversaturated, and the pore saturation is not in a stable equilibrium. On measurement day 52, the embedded sensors of C1 and C3 are leaving saturation. On day 77, C2 does the same. Due to further evaporation and hydration, the moisture content and the relative humidity continue to decrease. Eventually, the corresponding relative humidity is between 80.8% RH and 83.5% RH and would convert towards the ambient humidity of 50% RH afterwards. In general, the relative humidity curves of the three tested cement pastes are similar in shape and amplitude. The results show that the embedded HF sensors might be used for quality control during the construction process to monitor cement hydration. UHF sensors were embedded in this preliminary study as well. Although communication with the UHF sensors was possible, humidity data were never received. The moist concrete leads to a significant attenuation of the electromagnetic field at the UHF frequency. Thus, the required supply voltage of the humidity sensors was probably never reached.

### 4.2. Long-Term Measurement from RFID-Based Sensors Embedded in Building Materials

Based on indications of decomposition of the casings (further results in Section 4.3), for long-term study, the casing material was changed to polyvinyl chloride (PVC) (Figure 3 and Figure 4). The optimised sensors were embedded in another test specimen. This time, a cementitious screed was used instead of cement paste. Figure 8 shows the measured minimum transmitted power for HF- and UHF-based sensors embedded in the concrete specimen at 6 cm depth, according to [30]. As a first result, the transmitted power was measured for the HF and UHF system for more than 1000 days. Hence, the embedded systems are robust enough to work well and reliably in the high alkaline environment of building materials. Furthermore, the decreasing specimen weight is illustrated as well in Figure 8. In the first time period, the transmitted power of the UHF RFID is significantly reduced from 23.7 dBm to 19.1 dBm. In the course of this, the trend of the specimen weight and the UHF-transmitted power are similar. This correlation shows the attenuation of the UHF frequency of 868 MHz due to moisture. This frequency is in the same order of magnitude compared to dielectric moisture measurement tools such as microwave or radar. In the second and third measurement period, the UHF-transmitted power is increased to 29 dBm. Besides cross-influences from unknown sources, ageing of the circuit board or electronical components as well as degeneration of the antenna might have led to the increased energy consumption. The long interruption between the first and second measurement period was caused by technical problems. In comparison, the HF RFID based sensors require less energy on all measurement days. The last measurement point shows the highest value of 21 dBm. This slight increase after 950 days may indicate some ageing but could also be caused by an unknown cross-effect. Furthermore, the almost constant values in the first period indicate the independence of the HF-transmitted power from the material moisture. Both RFID systems were functional for more than 1000 days inside the concrete specimen and the reading of the transmitted power is still working, proving the principal suitability for long-term monitoring of concrete structures. 

Figure 9 presents the measured temperature and humidity values of the embedded RFID based sensor. For the UHF RFID based sensor, temperature and humidity data are recorded only for the first 156 days. After the long interruption of the measurements due to technical problems, the UHF sensor did not record humidity or temperature data anymore. After the first two days, the temperature remained stable at around 22.8 °C. The used mixing water was extracted directly from the water pipe without any heating. The lower temperature during the first two days is probably caused by the cold mixing water. Only five humidity values were recorded by the UHF sensor, decreasing from 100% RH to 76% RH. Within the first 20 days, the UHF-transmitted power was increased as shown in Figure 8. Thus, the energy supply was probably too low to power the humidity sensor. In the second and third measurement period, the UHF-transmitted power was even higher. Thus, the problem of insufficient power supply remained.

The HF RFID-based sensor recorded data all the time. The temperature was in the range of 21 °C most of the time. Until 156 days, the humidity sensor remained in saturation at 100% RH. The areas highlighted in pink indicate an ambient relative humidity above 70% RH (compare Figure 6). Nevertheless, on day 749, the corresponding relative humidity in the specimen was 91.5% RH and goes further down. However, on day 989, the sensor reached saturation again. This coincides with a time period having an ambient humidity above 70% RH and a drop of the ambient temperature to 16 °C. This temperature drop was well recognised by the sensor with a measured temperature of 15.7 °C. It cannot be evaluated if the concrete specimen really reached saturation again or if these values are a misinterpretation due to the volatile ambient conditions. However, after storing the specimen in another laboratory, the temperature increased, and the humidity decreased. In this condition, the specimen almost reached the equilibrium moisture content of 50% RH on measurement day 1016. The measurement values taken around day 989 clearly demonstrate that the HF RFID-based sensor still worked and had a high sensitivity to the changing ambient conditions. This emphasizes the capability of passive RFID based sensors for long-term monitoring. In case of a regular bridge inspection, such an unusual rise of humidity could be an indication for upcoming failure in the structural component. 

### 4.3. Long-Term Stability of Sensor Components and Materials in Alkaline Cementitious Environment

The RFID-based sensors must be protected against contact with the cement paste during concreting. If the cement paste reaches the circuit board, the alkaline environment would destroy the electronics. If the cement paste would reach the polymer-based membrane of the capacitive, relative humidity sensor element directly, the micro-particles of the cement paste would clog the polymer membrane. This leads to an irreversible shift in the total capacity of the polymer membrane and the measured capacity values would drastically deviate from the calibrated humidity values. 

Two measurement campaigns were carried out to prove the functionality and durability of the sensors. To check robustness, some sensors were extracted to obtain an impression of the condition of the sensor components and materials. In case of the preliminary study, the sensors were extracted from the concrete specimen after 180 days. In the second campaign, sensors were extracted from the screed specimen after three years. As can be seen on high-resolution photographs, epoxy resin clearly fails to withstand the alkaline environment. The PVC and the quartz glass filter membrane seem to be unaffected. A detailed discussion of the extracted sensors follows. 

Figure 10 depicts three setups of sensors casing after extraction from the concrete specimens. The sensors in Figure 10a remained in the specimens for half a year, the sensors in Figure 10b,c for almost three years. In Figure 10a,c), parts of the concrete are shown including fragments of the filter membranes. Figure 10a illustrates a RFID-based sensor, which was completely encapsulated in epoxy resin and remained in the specimen for half a year (coil antenna not shown in this photograph). During manufacturing and embedding, the epoxy resin was transparent with a smooth surface. After the extraction, the epoxy resin was no longer transparent. It has a dark brown colour indicating a chemical decomposition. Furthermore, the surface looks partially jagged and cracked. Although this is partly caused by the extraction process itself, we believe that swelling of the epoxy resin or shrinkage of the concrete has also led to the observed degeneration. Regarding the filter membrane made of quartz glass, discoloration at the surface is visible as well. Nevertheless, the extraction reveals that the filter remains unaffected inside. Therefore, the discoloration is probably caused by contact of the filter with the surrounding epoxy resin. At the fringe of the filter one might get the impression that the epoxy resin is entering the filter membrane due to capillary suction. If the entire membrane would be penetrated and sealed with epoxy resin, no exchange of the relative humidity could occur. Eventually, epoxy resin is considered as inappropriate for sensor casings in high-alkaline building materials. 

Figure 10b shows the RFID-based sensors in a PVC casing which was embedded for almost three years. Additionally, the circuit board was encapsulated in epoxy resin. The PVC casing looks completely unaffected. The epoxy resin inside the sensor had no direct contact to the concrete. Nevertheless, discoloration occurred, and the surface appears jagged again. Cracks are not visible so far. The white areas around the circuit board are fragments of the displaced filter membrane, which was fixed with ordinary hot glue. The hot glue is still transparent. Based on the observations in Figure 10a,b, the discoloration of the epoxy resin might be an ageing process which is independent of the alkaline construction material. Although we are not able to identify the cause of ageing definitively, the discoloration and jagged surfaces indicate a kind of chemical decomposition. Therefore, epoxy resin seems to be inappropriate for the encapsulation of embedded sensors, even if there is no direct contact with the alkaline building material. 

Figure 10c presents a RFID-based sensor covered only with a PVC casing after three years embedded in concrete. As seen before, the PVC casing is completely unaffected. Furthermore, the unshielded circuit board and the electronics do not show any signs of degeneration or ageing. Apart from some dust from the extraction process, the board and the components have the same appearance as during the embedment. Furthermore, the quartz glass filter is in excellent condition as well. Thus, a PVC casing and a membrane to protect the electronics against direct contact with the fresh concrete seems to be highly suitable for long-term operation. 

Besides the casing materials, the electrical components must withstand the alkaline environment. One crucial part is the relative humidity sensor element. This sensing element relies on the relative humidity exchange between the porous material and the air-filled void in the sensor, as shown in Figure 1. Thus, this sensing element with its polymer membrane is always exposed, which might reduce the long-time stability. Therefore, the condition after extraction of this single component is discussed in further detail.

The capacitive sensor element (HIH-5030, Honeywell (Morristown, NJ. USA)) for measuring the relative humidity is positioned directly on the circuit board. This component has an opening, i.e., orifice, to enable convection of the relative humidity towards the polymer-based membrane. The casing of the humidity sensor element has a cross section of 4 mm × 8 mm. The diameter of the orifice is approximately 1.2 mm. Detail photographs are shown in Figure 11. Figure 11a shows the orifice of polymer membrane of the sensor shown in Figure 10a. In this case, the entire RFID-based sensor was encapsulated by epoxy resin and remained in the concrete specimen for half a year. The polymer membrane shows degeneration and ageing like the epoxy resin (red arrows). It seems that decomposed parts of epoxy resin have entered the humidity sensor and partly cover the membrane. If the entire membrane is affected, the sensor could not interact with the surrounding air anymore. In case of partial penetration, the total capacity has probably shifted, and the calibration function does not represent the true humidity anymore. In Figure 11b, the sensor was encapsulated by epoxy resin again and remained in the concrete for half a year (similar to the sensor in Figure 10a). In this case, no traces of epoxy resin are visible. Nevertheless, the polymer membrane shows distinct areas of degeneration (red arrow). The surface has become rougher and it seems that the polymer is partially dissolved. Furthermore, a substance resembling white crystals is present around the orifice and the polymer membrane. These crystals, although in reduced size and quantity, are also visible in Figure 11a. In both cases (a) and (b), an encapsulation made only by epoxy resin was not able to protect the humidity sensor elements and the polymer membrane from degeneration. 

Figure 11c shows the orifice and the polymer membrane of sensor Figure 10b, which was embedded for almost three years. No decolouration indicating degenerated epoxy resin is visible. Only a small area with a diameter of approximately 0.1 mm looks different (red arrow in Figure 11c). Based on this photograph, it is not clear if the polymer membrane has degenerated or if only a small dust particle lies on the membrane. In general, the humidity sensor appears to be in good condition. Figure 11d depicts the sensor of Figure 10c, which was only encapsulated by the PVC casing and remained in the specimen for almost three years. The polymer membrane and all the surroundings are in excellent condition. Based on these four detailed photographs of the orifice, one has to conclude that epoxy resin has a negative effect on the sensor and the polymer membrane. In case of encapsulation purely in epoxy resin, significant damage of the sensor is documented in Figure 11a,b. On the other hand, in total absence of epoxy resin, the polymer membrane has the best condition (Figure 11d). Thus, a sealed PVC casing seems to be the most promising approach for long-term durability and functionality. 

### 4.4. Accelerated Aging Test of the Casing and Filter Membrane Materials

The quartz glass filter as well as the PVC appeared to be in good condition after almost three years embedded in concrete. However, to prove the stability of the materials on the micro-scale, further tests were required [29]. All materials were tested in an SH solution (pH-value of 12.7) for 12 days. The weight was measured to find out if the materials showed a mass loss due to chemical reactions. As depicted in Table 3, the epoxy resin showed a small mass loss of approximately 0.1%. In contrast to current measurements, older tests in SH solution yielded mass increase of epoxy resin of 1.79% [29]. Independent of a mass increase or mass decrease, any kind of mass variation indicates an interaction of the epoxy resin with the alkaline environment. Epoxy resin cannot be considered as inert in this setup. Thus, epoxy resin is considered as unsuitable for long-term stability in high alkaline environment. Another tested material is PVC. Although PVC is not a pure hydrocarbon chain, it is often used in chemical facilities and for waste water pipes. Thus, in practice, PVC withstands aggressive and alkaline environments. Furthermore, PVC parts are easy to glue together unlike pure hydrocarbon materials like polyethylene. However, the tests reveal that PVC remains unaffected by the alkaline environment. The small mass increase of 0.02% was considered as being a measurement uncertainty. These results are in line with chemical resistance charts of polymers.

The quartz glass filter with a modal pore size of 12 µm and a thickness of 4 mm shows a mass increase of 0.88% within the first two days, followed by a mass loss up to −2.22%. The temporal trend and the corresponding mass variations are in excellent alignment to older measurements [29]. Thus, quartz glass filters do not withstand the high alkaline environment. Moreover, droplet tests on the surface were carried out. The investigated quartz glass filter showed a pronounced tendency for capillary suction. This leads to liquid water transport through the pore system to the opposite side of the membrane. If the membrane is positioned horizontally, a droplet meniscus is formed. If this meniscus comes into contact with any surface inside the sensors casing, the entire volume might be filled with water. Therefore, the investigated quartz glass filters were insufficient to protect the RFID-based sensors against fresh cement paste and for long-term application in concrete. 

However, polytetrafluoroethylene (PTFE) was found to be an alternative. The PTFE filter shows a highly hydrophobic behaviour (Permeaflon – Berghof Fluoroplastics (Eningen, Germany)) Water droplets remain on the surface and do not penetrate the filter membrane. This alternative will be further investigated and is likely to be used in the next optimization step of the RFID-based sensor. 

## 5. Discussion

Infrastructure made of reinforced concrete has an expected lifespan between 50 and 80 years or even more. Most damage and degeneration occur during the last third of the lifespan. Hence, embedded sensors must be able to provide reliable measurement data for several decades. Passive embedded sensors yield a promising opportunity for long-term monitoring of structural components in civil engineering. As demonstrated, the RFID-based sensors used, which were embedded in concrete, are able to deliver measurement values for more than 1000 days. This long-term study shows that the concept of passive sensors for life cycle monitoring is promising. 

For more holistic monitoring, more critical parameters should be measured. For reinforced concrete, this would be, among others: electrochemical potential, chloride penetration, O_2_ and CO_2_ concentration, electrolytic resistance, or the pH-value. All these values can be useful for predictive maintenance and can contribute to do a reasonable service life estimation. It would be favourable, if one RFID-based sensor would be able to record all these parameters. The RFID chip, used in the study, had only one analogue input channel. This is a strong limitation for complex setups. The next generation of sensors, which are currently being developed, will use digital sensors instead. With a low-energy micro-controller, several parameters can be measured at the same time. It is also possible to install similar, redundant sensors in order to reduce the measurement uncertainty.

The further developed sensor systems include six components and were extended in their possibilities; the sensor system consists of the components which were taken over from the previous version of the sensor systems and newly added components. These include a digital temperature and humidity sensor (ENS210), a low-power microcontroller (PIC16(L)F1516 with nanoWatt XLP Technology) with necessary interfaces, digital I/O, I2C, SPI and ADC, to establish communication between the components, to receive data, or to supply the external sensors, and an analogue adaptation circuit for analogue sensors, e.g., to measure the electrochemical potential. The antenna is connected to the RFID IC by appropriate antenna matching. In order to keep the design universal, we have developed a uniform setup which should only be adapted to the pin assignment of the RFID IC (asm SL13A for HF and SL900A for UHF). Figure 12 shows the block diagram consisting of the basic components, which connect the sensors, and the RFID components with the antenna, the latter being the interface to the user.

The current RFID transponders are still in the validation phase. The current prototype is shown in Figure 13. It is planned to connect several sensors as a network in order to cover a larger area with one RFID transponder. The sensors are addressed either with the digital or analogue interface: up to 5 analogue sensors and, additionally, a number of digital sensors, depending on the number of available pins. The warming of the circuit board might have an influence on the measured temperature value of the sensor. Thus, a cut around the temperature sensor is recommended. Further connection points are available for the connection of more sensors or to allow an adjustment of the circuit.

The change from analogue to digital sensors will also lead to lower energy consumption, which is an overall purpose. The lower the energy consumption, the more complex RFID-based sensor solutions can be. Furthermore, the sensors do not have to be installed directly on the circuit board. Sensors could be locally distributed and connected via cables to the RFID. All cables would be embedded as well. In this setup, the RFID chip would act as a kind of hub. The energy is distributed from the RFID hub to the individual sensors and data are collected. For all these imaginable setups, the energy consumption is the limiting factor. There are three general approaches: increasing the power of the transmitter, increasing the efficiency of the RFID antenna, decreasing the power consumption of all electric components of the embedded sensor.

A second benefit of low demand losses is the increase in the reading range. Passive systems are limited. Passive HF RFID-based sensors have maximum working ranges of 0.1 m to 0.3 m and UHF RFID based sensors of 0.3 m to 2 m. The working range in this context means that the installed sensors receive enough energy for data recording. If only the chip ID is of interest, the reading range might be higher. In almost all cases, the first layer of reinforcing bars is the most sensitive zone in reinforced concrete. Their depth is determined by the concrete cover which usually ranges from 3 cm to 7 cm. Nevertheless, long reading ranges of several meters would make the RFID read out significantly easier. For example, foundations or the bottom side of bridge decks are difficult to access. A long-range read out from the bridge deck would be a great advantage for bridge monitoring. 

As a last step, a system design, planned for being in function for decades, must be highly compatible for future communication terms. The great advantage of HF RFID-based sensors is the communication frequency of 13.56 MHz. It is the same frequency that is used by near field communication (NFC). The technology is a standard element in every modern smartphone and tablet. Therefore, the sensors could be read out by an ordinary smartphone app. This would increase the acceptance and feasibility in the civil engineering branch. Only robust, long-life, and easy-to-use systems will have a chance in the market. 

## 6. Conclusions

In civil engineering, structures such as bridges or tunnels have a service life of several decades. To cover this long time period and to enable the required robustness against the rough environment, passive RFID-based sensors are embedded into concrete. The materials are tested to withstand the high alkaline environment. Suitable membrane materials are found to facilitate the measurement of the corresponding relative humidity of the porous cement matrix. By embedding the passive RFID-based sensors in different concrete mixtures, the hydration of the cement matrix was monitored for 170 days. Furthermore, long-term tests for more than 1000 days were successfully performed. The measured temperature and relative humidity were sensitive to changes of the ambient condition. Thus, this setup is considered as applicable for structural health monitoring in civil engineering. Moisture monitoring can reveal unexpected moisture increases and early repair interventions can be carried out before cost-intensive degradations occur, e.g., corrosion. This approach might contribute to a significant reduction of the overall repair costs in the future. 

## Figures and Tables

**Figure 1 sensors-19-05514-f001:**
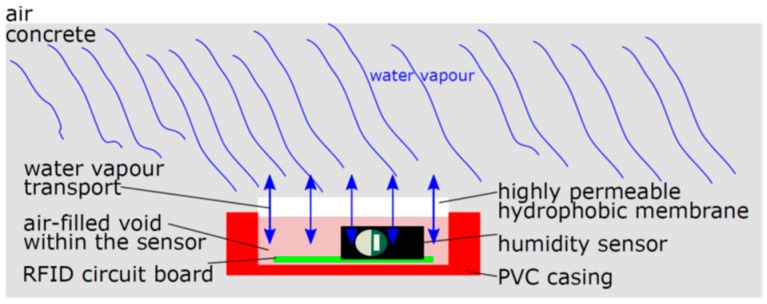
Schematic sketch of the system for measuring the corresponding relative humidity with an embedded radio frequency identification (RFID)-based sensor.

**Figure 2 sensors-19-05514-f002:**
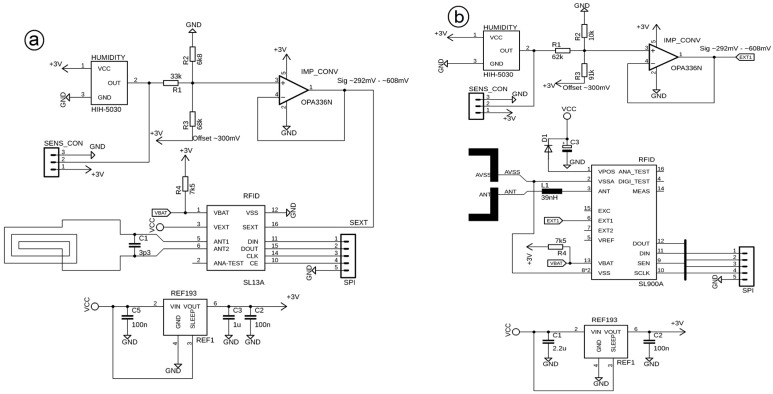
Circuit of the currently embedded sensor system with matching details: (**a**) high frequency (HF), (**b**) ultra-high frequency (UHF).

**Figure 3 sensors-19-05514-f003:**
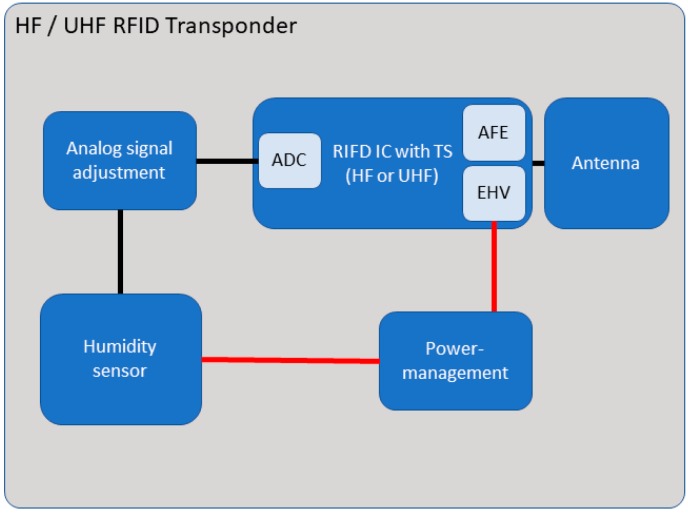
Block diagram of the currently embedded sensor system.

**Figure 4 sensors-19-05514-f004:**
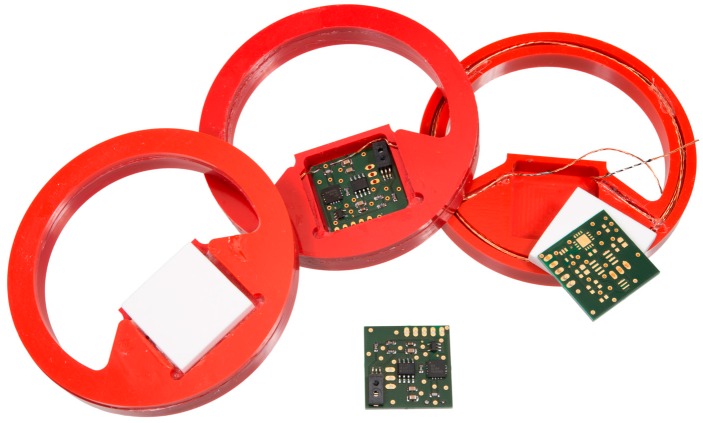
Photograph of the embedded HF RFID-based sensors (with PVC casing) for measuring the corresponding relative humidity.

**Figure 5 sensors-19-05514-f005:**
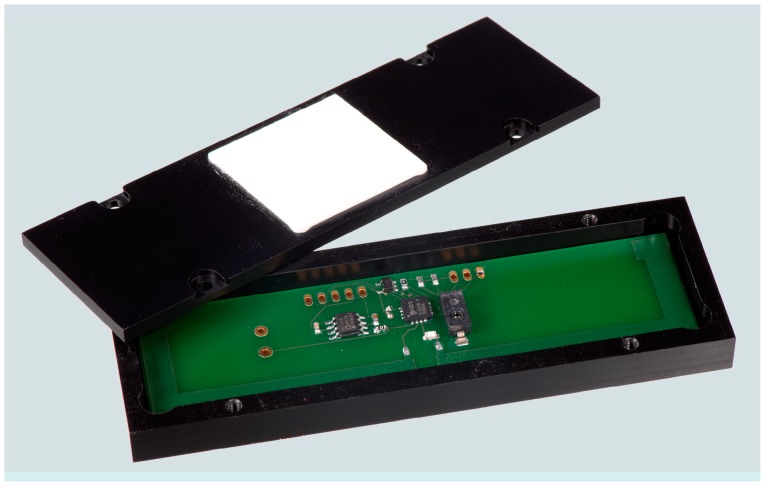
Photograph of the embedded UHF RFID-based sensors (with PVC casing) for measuring the corresponding relative humidity.

**Figure 6 sensors-19-05514-f006:**
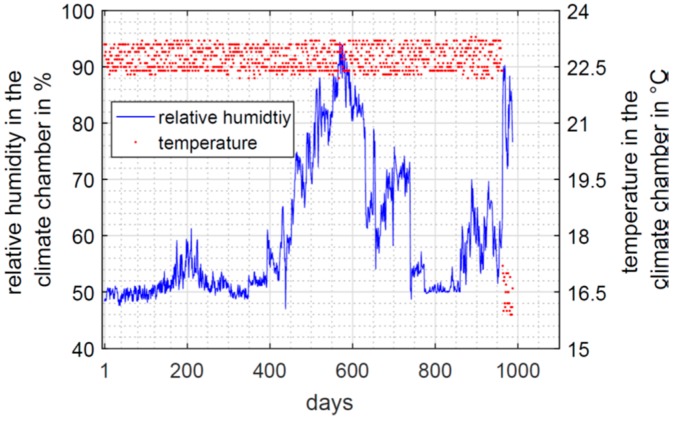
Ambient relative humidity and ambient temperature in the climate chamber.

**Figure 7 sensors-19-05514-f007:**
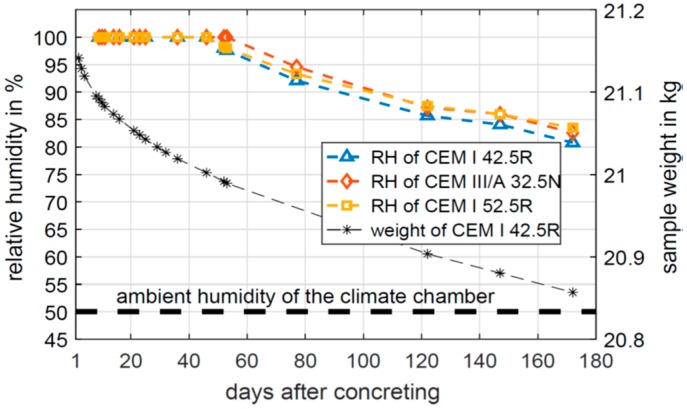
Evolution of the corresponding relative humidity and the specimen weight at 3 cm depth.

**Figure 8 sensors-19-05514-f008:**
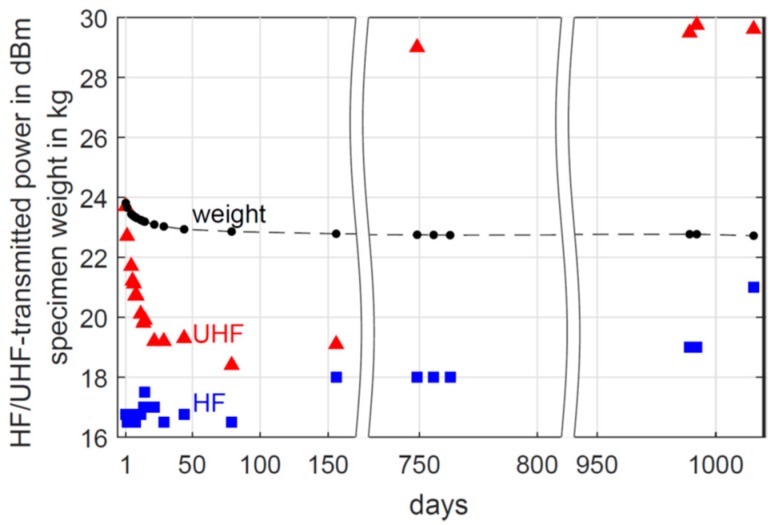
Transmitted power of the embedded HF and UHF RFID-based sensor and specimenweight.

**Figure 9 sensors-19-05514-f009:**
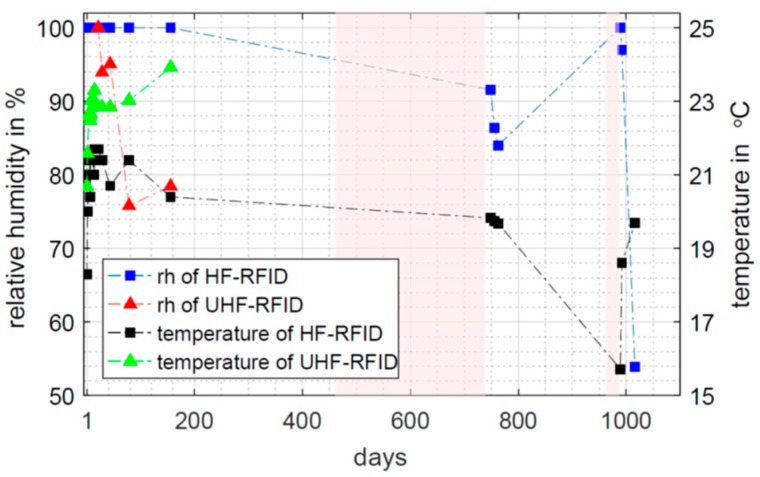
Corresponding relative humidity and temperature at 6 cm depth measured with an embedded HF and an UHF RFID sensor; areas highlighted in pink indicate mark time periods with ambient humidity above 70% RH.

**Figure 10 sensors-19-05514-f010:**
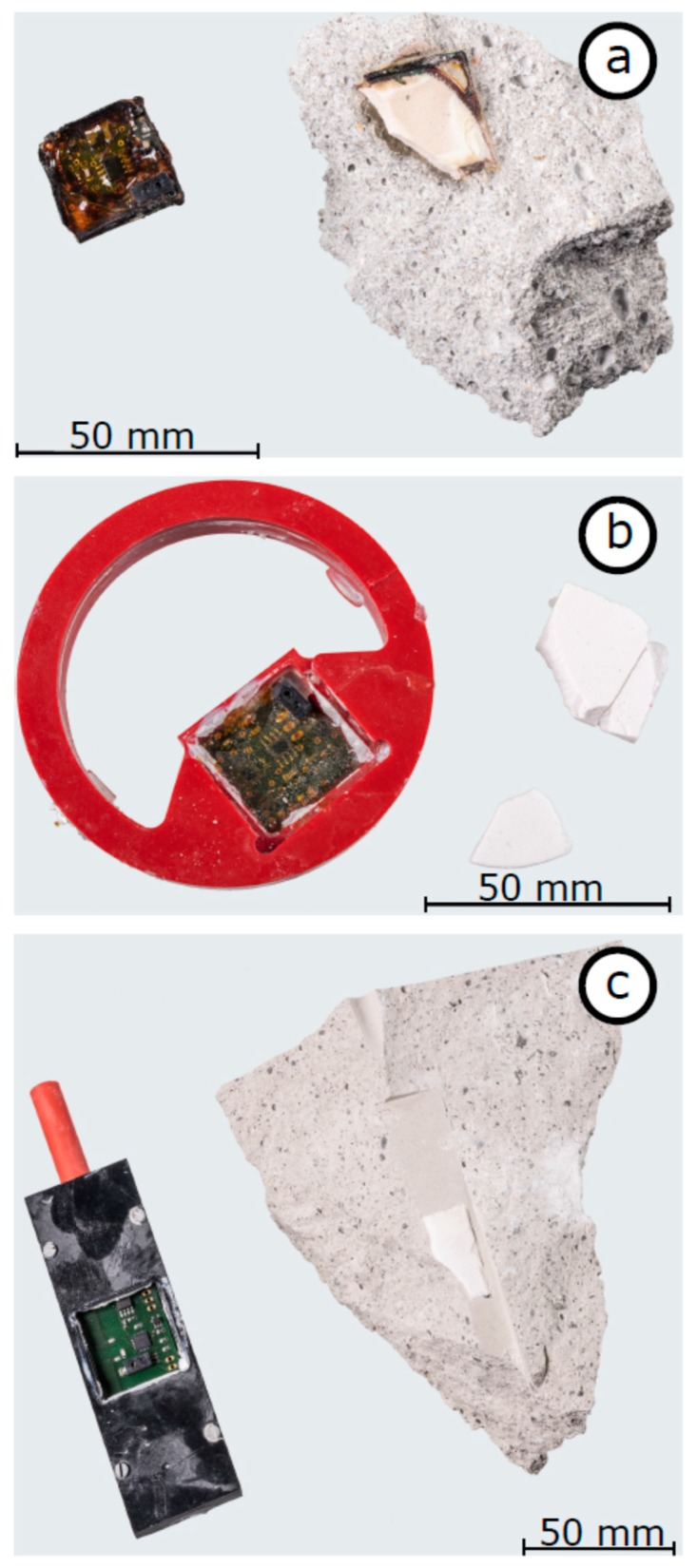
Photographs of the extracted sensors after being embedded into concrete; (**a**) epoxy resin casing extracted after 180 days; (**b**) PVC casing and additional epoxy resin cover on the circuit board extracted after 3 years; (**c**) PVC casing extracted after 3 years.

**Figure 11 sensors-19-05514-f011:**
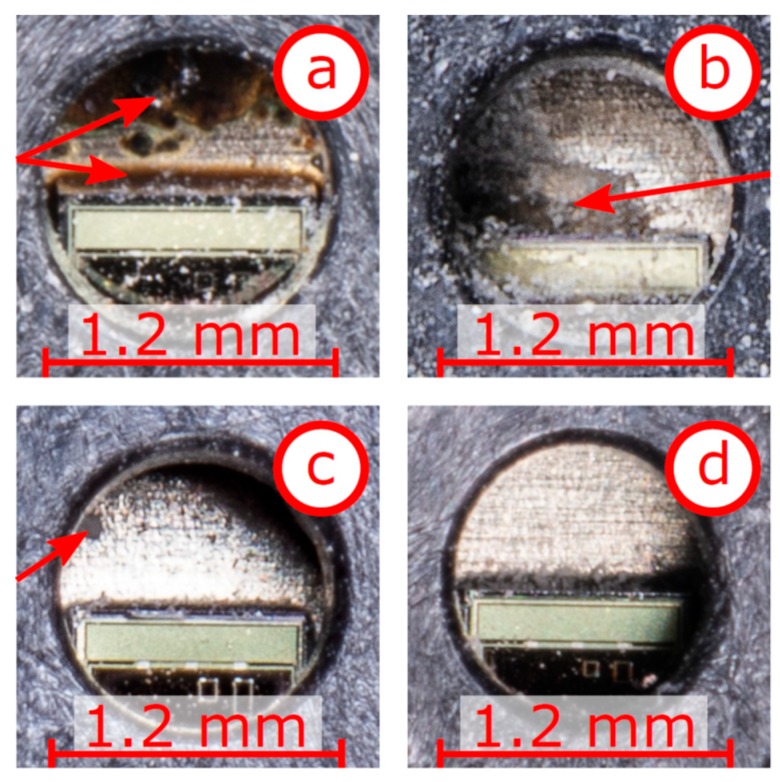
Detailed photographs of the orifice of the capacitive humidity sensor elements (approximately 1.2 mm in diameter) with the underlying polymer membrane; (**a**,**b**) sensors with epoxy resin casing embedded for half a year; (**c**) with PVC casing and additional epoxy resin cover on the circuit board embedded for almost three years; (**d**) with PVC casing embedded for almost three years.

**Figure 12 sensors-19-05514-f012:**
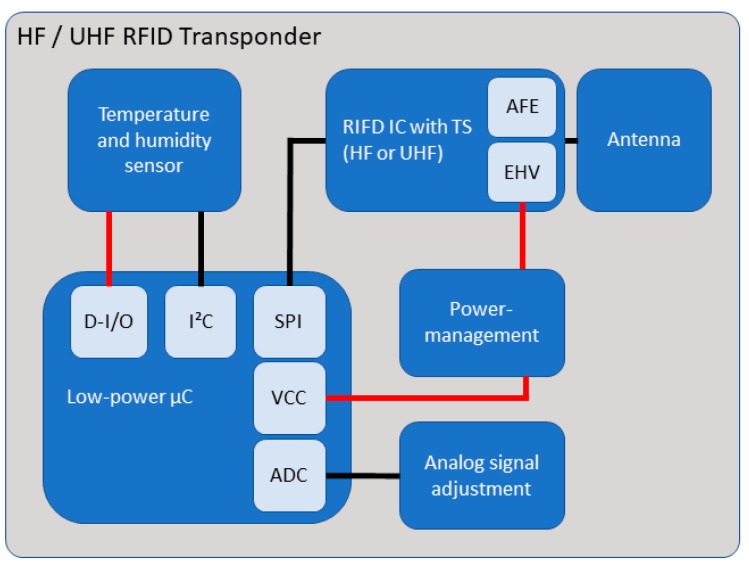
Block diagram of the new digital sensor systems.

**Figure 13 sensors-19-05514-f013:**
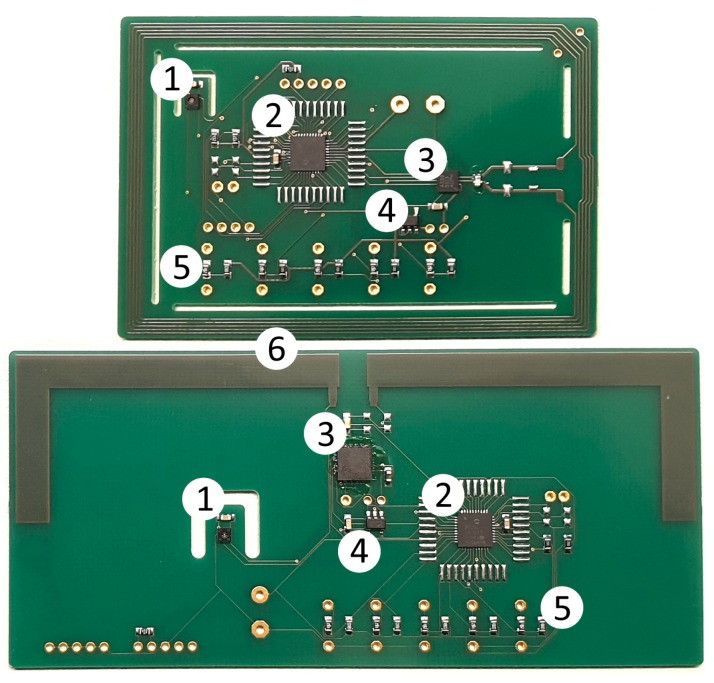
New RFID-based sensors: (1) temperature and humidity sensor; (2) low-power microcontroller; (3) RFID IC; (4) external power management; (5) analogue connections with matching circuit for sensors; (6) antennas.

**Table 1 sensors-19-05514-t001:** Overview of the different measurement campaigns.

Section	Title	Purpose	Considered Quantities
4.1	Preliminary case study	Initial prove of the measurement concept	Relative humidity
4.2	Long-term measurements	Verification of the sensor long-term stability	Transmitted power Relative humidity Temperature
4.3	Examination of embedded components and materials	Investigation of the components exposed to concrete	Visual inspection
4.4	Accelerated aging tests of the used materials	Laboratory tests in sodium hydroxide (SH) solution for detailed material characterisation	Weight of material samples

**Table 2 sensors-19-05514-t002:** Parameter of the used cement paste.

Concrete Type	Abbreviation in the Text	Compressive Strength (Nmm^−2^)	Water to Cement Ratio	Hydration Type
CEM I 42.5R	C1	42.5	0.5	rapid
CEM III/A 32.5N	C2	32.5	0.5	normal
CEM I 52.5R	C3	52.5	0.4	rapid

**Table 3 sensors-19-05514-t003:** Mass variation of the tested casing materials in a 1 mol. SH solution with a mean pH-value of 12.7.

Material	Mass Variation after 2 Days in %	Mass Variation after 5 Days in %	Mass Variation after 7 Days in %	Mass Variation after 12 Days in %
Epoxy resin	−0.02	−0.09	−0.12	−0.07
PVC	0	0	0	+0.02
Quartz glass filter	+0.88	−0.05	−1.54	−2.22
PTFE 1 µm	0	−0.02	−0.06	−0.08
PTFE 10 µm PTFE 50 µm	0 0	0 0	0 0	−0.01 0
PTFE 100 µm	0	+0.01	+0.01	+0.01

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
