# Peer review of "Smart RFID Sensors Embedded in Building Structures for Early Damage Detection and Long-Term Monitoring"

_sensors, 2019, doi:10.3390/s19245514_

Round 1

Reviewer 1 Report

This paper reports the experimental results of short-term and long-term performance evaluation of high frequency (HF) and ultra high frequency (UHF) RFID sensors embedded in concrete. The authors provided detailed moisture monitoring data as well as discussions for actual implementation, so this reviewer consider this paper as a valuable contribution to the wireless sensing field. However, this manuscript requires to be significantly improved in editorial point of view: language, broken reference links, ineffective description of results, and so on. Therefore, this reviewer recommends to accept after revisions done. The detailed comments are written below:

Major points:

Major revision by native English speaker may be required. All references to figures or tables, and citations are broken. Check your PDF before submitting the manuscript Figures 5, 6, 7 are not directly mentioned in the main texts, although they are the main results. And the description of those figures are lengthy and not organized. Please articulate and organize better way. 2. This results section should be re-written to be more concise, well-organized. If necessary, add one more figure and explain better. In Figure 7, the most points are done during 1-160 days, and once on 750 day, and three times around 1000 day. How did you design this measurement interval? To be objective, equi-interval (maybe every 100 days?) would be great after 160 days. Please add justification.

Minor points:

Line 54: the Bundesanstalt fur Materialforschung und -prufung Figure 4 need dimension P6 Line 212: The transmit (or transmission) power steps are 0.25 dBm … (for both HF and UHF?) Line 237: a NaOH solution – use the name ‘Sodium Hydroxide (SH)’ Figure 10: Use higher resolution photo.

Author Response

Major revision by native English speaker may be required.

Answer: The document was already checked by a professional editor. She is a native speaker from Bosten, MA. Nevertheless, a colleague checked abstract, chapter 1 and 2 and the conclusion again. He is a native speaker from Texas.

All references to figures or tables, and citations are broken. Check your PDF before submitting the manuscript

Answer: In our submitted version, word and pdf, all figures and references were linked dynamically. Furthermore, in the actual word and pdf version which was already edit by MDPI, all links still work in pdf and word (we had to download this version from the MDPI-homepage). We are supposed to use this version for the revision. As I said, all links still work, and we are directed to the figure/reference by clicking the number in the text. Unfortunately, we do not know what kind for document you received, but in all documents available for us, the links work appropriately.

Figures 5, 6, 7 are not directly mentioned in the main texts, although they are the main results.

Answer: I rechecked the word and the pdf downloaded from the MDPI homepage after the reviews. All tables and figures are mentioned and linked in the text. The link in the text is always given in the paragraph before the figure appears (as required by the journal). We believe this a consequence of the issue answered above. Unfortunately, we do not know what kind for document you received, but in all documents available for us, the figures and tables are named and linked in the text appropriately.

And the description of those figures are lengthy and not organized. Please articulate and organize better way.

Answer: We shortened caption of figure 6, 7, 9. Please see the word with track changes

This results section should be re-written to be more concise, well-organized. If necessary, add one more figure and explain better.

Answer: We put a table as overview on top of section 4. We adjusted figure 7. We shortened the text to give more focus on the long-term stability. Please see the several changes in the text, especially chapter 4.

In Figure 7, the most points are done during 1-160 days, and once on 750 day, and three times around 1000 day. How did you design this measurement interval? To be objective, equi-interval (maybe every 100 days?) would be great after 160 days. Please add justification.

Answer: Indeed, more measurement values and a more equidistant distribution of the values would be favorable. However, all was work in progress and we had to fix some very time-consuming hardware and software problems. Added to the text: The long interruption between the first and second measurement period was caused by technical problems.

Minor points:

Line 54: the Bundesanstalt fur Materialforschung und -prufung

Answer: We are forced by our institute to use the German name in the text. We put it in quotation marks.

Figure 4 need dimension P6 Line 212:

Answer: We believe you meant figure 5 (figure 4 has no units). Missing dimension added to figure 5.

The transmit (or transmission) power steps are 0.25 dBm … (for both HF and UHF?)

Answer: Text was added: The minimum transmit power step was selected for the two frequencies, i.e. 0.25 dBm steps for HF and 0.1 dBm steps for UHF.

Line 237: a NaOH solution – use the name ‘Sodium Hydroxide (SH)’

Answer: We changed it to sodium hydroxide (nine times in total in the text)

Figure 10: Use higher resolution photo.

Answer: Each subfigure (a,b,c,d) of approximately 3.5 times 3.5 cm has 450 to 450 pixels, resulting in a resolution of around 330 dpi. This should be sufficient, maybe your version of the document is somehow downscaled?

Reviewer 2 Report

The main goal of the revised manuscript is a wireless RFID based sensors, which are embedded completely in concrete. The paper describes the measurement principle and the system design of an exemplary passive RFID-based sensor and presents comprehensive results of validation experiments, especially regarding the chances and challenges for long-term application. The principle of measuring the equilibrium moisture content based on embedded relative humidity sensors, the so called corresponding relative humidity, is described. Then, the adaption of the sensor to the RFID chip for communication and energy supply is discussed. At the end, the used RFID based sensors are shown.

That being said, the current paper suffers, in my opinion, from some imprecision that should be corrected: in particular, the paper should be better organized and should give more details about the sensor (electrical schematic, components, material ect). In my opinion the paragraph on the results is unclear and very confusing.

From the aforementioned considerations, I don’t recommend for publication of the submitted paper but I encourage the authors to make changes and resubmit it.

In the next some general comments:

In the Introduction the authors should explain better what are the innovations introduced in their design compared to the state of the art. Moreover, in the references, several articles written by the authors deal with the same themes. Authors should indicate what's new in this paper compared to previous ones.

I found a paper in 2017 IEEE SENSORS entitled “Embedded passive RFID-based sensors for moisture monitoring in concrete” that has parts in common with this. In my opinion the authors should better explain the differences/additions between the two.

The paper describes an HF and VHF sensor, but only the block diagram is shown. In my opinion, a section should be added with a detailed description of the sensor and its operation from an electrical point of view.

Specific comments

In section 3.2, in my opinion, the authors should add a figure of the structure to better understand how the sensor is positioned inside the wood.

In paragraph 3.3 Fig.5, in my opinion, the authors should give an explanation on the variations measured around day 200.

In paragraph 3.3 row 232-234, the authors test the water vapour permeability of the humidity sensors, a smaller, ventilated climate chamber. Where are the results?

In paragraph 4.1 the authors test the HF and VHF RFID sensors in different types of concrete but show the results only for the HF sensor. In my opinion, the results of the VHF tag sensor should also be added.

In paragraph 4.2, the authors show the long-term measurement data (about 1000 days) of the embedded sensors only for the first 150 days or so and for the last 250 days. In my opinion, the results should be shown at fairly regular intervals. If it is not possible, the authors should give an explanation.

In paragraph 4.2 row 313-315, I didn't understand why the UHF sensors were left alone until day 156? In paragraph 4.3 it is said instead that the sensor remained 3 years in the concrete.

In paragraph 4.2 row 329, why are the humidity values different from those measured in the samples in section 4.1?

Pag. 4 row 146-155: In my opinion, this part, quite well known in the scientific community about RFID, should be shortened.

In Fig. 5, the y-axis label is incomplete.

Fig. 7 and Fig 8 are not clear, in particular the first part of the measurement data is too close, in my opinion, it would be better to add next to another figure with a zoom.

Fig. 10: in my opinion, a photo of a new sensor should be added, this way it is easier to understand how the sensor itself is made.

Author Response

That being said, the current paper suffers, in my opinion, from some imprecision that should be corrected: in particular, the paper should be better organized and should give more details about the sensor (electrical schematic, components, material ect).

Answer: We added several details to the used electrical components and systems. Please see entire text, especially section 2.2. Regarding electrical schematics, in our opinion, it would change the focus of the paper too much. The main focus is the long-term stability of the already developed RFID systems into building materials. Nevertheless, we added two more references which discuss the electrical schematics and boards in detail.

In my opinion the paragraph on the results is unclear and very confusing.

Answer: We put a table as overview on top of section 4. We adjusted figure 7. We shortened the text to give more focus on the long-term stability. Please see the several changes in the text.

From the aforementioned considerations, I don’t recommend for publication of the submitted paper but I encourage the authors to make changes and resubmit it.

In the next some general comments:

In the Introduction the authors should explain better what are the innovations introduced in their design compared to the state of the art. Moreover, in the references, several articles written by the authors deal with the same themes. Authors should indicate what's new in this paper compared to previous ones.

I found a paper in 2017 IEEE SENSORS entitled “Embedded passive RFID-based sensors for moisture monitoring in concrete” that has parts in common with this. In my opinion the authors should better explain the differences/additions between the two.

Answer: The IEEE publication deals only with the first results, which is comparable with section 4.1. Several additional results are added now, especially the long-term measurements and the material characterization of the extracted sensors. We added the IEEE publication (I though we already had it in our references) and added text at the end of section 1.

The paper describes an HF and VHF sensor, but only the block diagram is shown. In my opinion, a section should be added with a detailed description of the sensor and its operation from an electrical point of view.

Answer: In our opinion, this comment is fairly similar to your first comment. We apologies, but we do not have further ideas. Thus, we repeat our answer from our first comment. We added several details to the used electrical components and systems. Please see entire text, especially section 2.2. Regarding electrical schematics, in our opinion, it would change the focus of the paper too much. The main focus is the long-term stability of the already developed RFID systems into building materials. Nevertheless, we added two more references which discuss the electrical schematics and boards in detail.

Specific comments

In section 3.2, in my opinion, the authors should add a figure of the structure to better understand how the sensor is positioned inside the wood.

Answer: We added text and a reference which includes a photograph of the wooden template (and more measurement results). We believe a complete figure is too much repetition of already published work.

In paragraph 3.3 Fig.5, in my opinion, the authors should give an explanation on the variations measured around day 200.

Answer: Text was added, please see the word file with track changes.

In paragraph 3.3 row 232-234, the authors test the water vapour permeability of the humidity sensors, a smaller, ventilated climate chamber. Where are the results? Answer: Text was added, please see the word file with track changes.

 In paragraph 4.1 the authors test the HF and VHF RFID sensors in different types of concrete but show the results only for the HF sensor. In my opinion, the results of the VHF tag sensor should also be added.

Answer: We never received humidity data of the UHF tag. Text was added, please see the word file with track changes.

In paragraph 4.2, the authors show the long-term measurement data (about 1000 days) of the embedded sensors only for the first 150 days or so and for the last 250 days. In my opinion, the results should be shown at fairly regular intervals. If it is not possible, the authors should give an explanation.

Answer: Indeed, more measurement values and a more equidistant distribution of the values would be favorable. However, all was work in progress and we had to fix some very time-consuming hardware and software problems. Added to the text: The long interruption between the first and second measurement period was caused by technical problems.

In paragraph 4.2 row 313-315, I didn't understand why the UHF sensors were left alone until day 156? In paragraph 4.3 it is said instead that the sensor remained 3 years in the concrete.

Answer: Indeed, the HF and the UHF sensors remained in the sample for three years. And were able to read out the ID of the HF and the UHF sensor all the three years. But we got only 5 humidity measurements with the UHF sensors, see figure 8. Based on figure 7, the minimum transmitted power must be approximately under 19 dBm for humidity measurements. In short, the sensors were embedded for 3 years but the UHF delivered only values from day 22 to day 156. We modified the text to empathizes this issue.

In paragraph 4.2 row 329, why are the humidity values different from those measured in the samples in section 4.1?

Answer: In section 4.1, we describe results of preliminary tests. In section 4.2, we show the long-term measurements of the new and optimized sensors in a new mixture. The mixture changed from cement paste to screed. Furthermore, the depth changed from 3 cm to 6 cm. Text was added, please see the word file with track changes.

Pag. 4 row 146-155: In my opinion, this part, quite well known in the scientific community about RFID, should be shortened.

Answer: We shortened this paragraph, please see the word file with track changes.

In Fig. 5, the y-axis label is incomplete.

Answer: Missing dimension added to figure 5.

Fig. 7 and Fig 8 are not clear, in particular the first part of the measurement data is too close, in my opinion, it would be better to add next to another figure with a zoom.

Answer: We modified figure 7. The measurement values are now easy to identify. We though a lot about figure 8. At the end, we decided to remain with figure 8. The single values in the beginning are of minor interest. The main issue is the high humidity and temperature deviation after day 700. The climate chamber regulation was broken, and we had “crazy” ambient conditions. However, the RFID sensors we able to follow the trend after three years, and this is the main message. We shortened the text, especially the description of the first days. We hope that the text and the figure harmonise better now.

Fig. 10: in my opinion, a photo of a new sensor should be added, this way it is easier to understand how the sensor itself is made.

Answer: We do not modify/develop the relative humidity sensor itself. This is a commercial one. We added some text to make this clear, please see the word file with track changes.

Reviewer 3 Report

This paper presents RFID sensing platforms embedded in building structures for humidity monitoring. In my opinion, there are several major issues that need to be considered prior to its publication:

1) Are the RFID platforms in Figs. 3 and 4 for HF and UHF band commercial platforms? They seem to be commercial. Which ones? 

2) According to the manufacturer specifications, the Scemtec SAT-A4-LR-P transmitter is meant for 125 kHz. How is it possible that the authors used it for HF and UHF systems?

3) Subsection titles are too long (for instance 3.3)

4) Correct typos such as "corrotion its initiated", "curciut", etc. "rh" should be in capital letters. Remove first sentence after "Author contributions"... In general, the paper gives the impression that it has not been carefully reviewed before the submission.

5) The most important issue is that authors are hiding critical information. For instance, information regarding the RFID chips used in their study or in the current RFID transponders in validation phase, antenna design and matching, the microcontroller used, etc. When you publish a paper, you need to give all the information to make the work reproducible. 

6) It is not clear if the current RFID transponders that are in validation phase have been used for any purpose or not. What is the point in showing an unfinished work?

Author Response

1) Are the RFID platforms in Figs. 3 and 4 for HF and UHF band commercial platforms? They seem to be commercial. Which ones? 

Answer: The main idea in this project was to use already existing RFID technology and adapt sensors to this system. Thus, we end up with RFID based sensing. This adaption of RFID to passive embedded sensors for civil engineering is completely developed by BAM. Both developments (HF and UHF RFID) are based on commercial components, the system with the corresponding circuit is a proprietary development of the BAM.

2) According to the manufacturer specifications, the Scemtec SAT-A4-LR-P transmitter is meant for 125 kHz. How is it possible that the authors used it for HF and UHF systems?

Answer: This was a typo, it must be Scemtec SAT-A4-LR-PP. It is the antenna and not a transmitter, for the measurement, two transmitters with the corresponding antenna have been used. Text was added in section 3.2.

3) Subsection titles are too long (for instance 3.3)

Answer: We shortened subsection title 3.3 and 4.3

4) Correct typos such as "corrotion its initiated", "curciut", etc. "rh" should be in capital letters. Remove first sentence after "Author contributions"... In general, the paper gives the impression that it has not been carefully reviewed before the submission.

Answer: We are not able to find the typo corrotion. Other referees stated that all links to figure are deleted. Maybe you received a version with an odd file-conversion. We changed circuit, RH, Author contributions. Furthermore, the entire document was checked by a professional native speaker from USA, Bosten, MA.

5) The most important issue is that authors are hiding critical information. For instance, information regarding the RFID chips used in their study or in the current RFID transponders in validation phase, antenna design and matching, the microcontroller used, etc. When you publish a paper, you need to give all the information to make the work reproducible.

Answer: We added several details to the used electrical components and systems. Please see entire text, especially section 2.2. Furthermore, we added two more references which discuss the electrical schematics and boards in detail.

6) It is not clear if the current RFID transponders that are in validation phase have been used for any purpose or not. What is the point in showing an unfinished work?

Answer: This research article describes RFID sensors at TRL 6 / 7. The objective and purpose of this development is given in the introduction. The ongoing long-term validation and optimization is a crucial aspect and will be continued over a longer period of time. BAM, as a federal research institute, is supposed to develop materials and techniques before commercialization. Our research findings should enable companies to finalize these products.

Round 2

Reviewer 2 Report

In my opinion, when a paper is presented, it must be clear and the reader must be able to reproduce what is presented. Here, even in the articles that are cited, it is not shown anywhere how the circuit is made specifically. This prevents anyone from reproducing this experiment.

From the aforementioned considerations, I don’t recommend for publication of the submitted paper but I encourage the authors to make changes and resubmit it.

Author Response

Comments and Suggestions for Authors

In my opinion, when a paper is presented, it must be clear and the reader must be able to reproduce what is presented. Here, even in the articles that are cited, it is not shown anywhere how the circuit is made specifically. This prevents anyone from reproducing this experiment.

From the aforementioned considerations, I don’t recommend for publication of the submitted paper but I encourage the authors to make changes and resubmit it.

Answer: Indeed, the circuits were part of the presentations, but they are not included in the cited papers. We would like to apologise for this confusion. We added the circuit of the HF and the UHF sensors as figures in the paper. Please see figure 2.

Reviewer 3 Report

- The authors have addressed my comments. although some revision is still required. In the added paragraph in Section 2.2, the following sentence makes no sense (it is not grammatically a sentence). Some commas or periods are missing:

One of these components is an analogue humidity sensor (Honeywell HIH-5030), which is connected to an internal RFID integrated circuit (IC) with temperature sensor (ams SL13A for HF and ams SL900A for UHF) analogue digital converter (ADC) via an analogue adaptation circuit, based on the superposition principle (Helmholtz), equation 1 has been adapted to the requirements.

- In the same paragraph, what is REF193? Again the sentence is not correctly written:

...which is regulated with the REF193 with the voltage regulators to 3 V in advance and the antenna with a matching of 3.3 pF at HF and 39 nH at UHF to analogue front end (AFE) as shown Figure 2.

Regarding the last sentence, authors should better explain the matching circuits, since they are different schemes for HF and UHF. 

Author Response

Comments and Suggestions for Authors

- The authors have addressed my comments. although some revision is still required. In the added paragraph in Section 2.2, the following sentence makes no sense (it is not grammatically a sentence). Some commas or periods are missing:

One of these components is an analogue humidity sensor (Honeywell HIH-5030), which is connected to an internal RFID integrated circuit (IC) with temperature sensor (ams SL13A for HF and ams SL900A for UHF) analogue digital converter (ADC) via an analogue adaptation circuit, based on the superposition principle (Helmholtz), equation 1 has been adapted to the requirements.

- In the same paragraph, what is REF193? Again the sentence is not correctly written:

...which is regulated with the REF193 with the voltage regulators to 3 V in advance and the antenna with a matching of 3.3 pF at HF and 39 nH at UHF to analogue front end (AFE) as shown Figure 2.

Answer: The entire Section 2.2 was modified and finally checked by a native speaker.

Regarding the last sentence, authors should better explain the matching circuits, since they are different schemes for HF and UHF. 

Answer: We added the electric circuits of the HF and UHF board. Please see the added figure 2. We believe this is a detailed explanation.
